# Revisiting Atomic Rounding with Vectorized Reparameterization for LLM Quantization

## Abstract

Large language models (LLMs) quantization predominantly relies on round-to-nearest (RTN) operations as the atomic operation to map floating point (FP) weights into quantization grids. Applied at tensor-, group-, or channel-level granularities, such non-element-wise rounding is sub-optimal, as it prevents error cancellation across elements. Adaptive rounding addresses this by assigning each weight an optimized rounding parameter, but existing methods introduce an auxiliary matrix of equal size to the weights, substantially inflating computation and memory costs. Thus, we propose VQRound, which re-parameterizes the rounding matrix via vector quantization (VQ) into a compact codebook, drastically reducing trainable variables while preserving quantization fidelity. We identify the critical role of the initialization of the rounding matrix, as a proper scheme minimizes the deviation from the FP model and facilitates efficient tuning of the rounding parameters. Beyond naive layer- or block-wise optimization, we introduce a lightweight end-to-end finetuning pipeline that requires only 128 samples and enables global optimization of codebooks across all layers. Moreover, VQRound can be used as a plug-and-play replacement for atomic rounding, complementing existing quantization techniques to further enhance accuracy. Experiments on billion-parameter models, including OPT, LLaMA, and Qwen, show that VQRound achieves competitive performance under 4-bit, 3-bit, or even 2-bit quantization with as few as 0.2% of the learnable parameters of prior adaptive rounding methods.

## 1 Introduction

Large language models (LLMs) have pushed the boundaries of natural language processing, but their rapid growth in parameters and context length comes with prohibitive compute and memory costs (Kwon et al., 2023). Low-bit weight quantization has emerged as a crucial tool to reduce model size and inference latency while preserving task performance, enabling more scalable and efficient deployment of LLMs. Existing methods such as GPTQ (Frantar et al., 2023) and QuaRot (Ashkboos et al., 2024) can achieve near lossless 4-bit quantization on large models. However, most quantization pipelines treat various components (*e.g.,* rounding strategy, grouping schemes, outlier handling) in isolation, which complicates their integration and limits overall efficiency.

A fundamental atomic operation in all quantization algorithms is rounding, which maps each floating-point weight to a nearby quantized value. The prevailing approach is round-to-nearest (RTN), which can be applied at the tensor, group, or channel level (Egiazarian et al., 2024; Frantar et al., 2023; Liu et al., 2025; Tseng et al., 2024). While simple, RTN independently rounds each weight and is theoretically suboptimal for minimizing global error. Nagel et al. (2020) pointed out that the task loss increase induced by quantization can be approximated by a quadratic form $\Delta \mathbf{w}^\top \cdot \mathbf{H}(\mathbf{w}) \cdot \Delta \mathbf{w}$. When $\mathbf{H}$ has non-zero off-diagonal entries, the quantization error of different weights interacts via cross terms $\Delta w_i \Delta w_j$. In such cases, the optimal rounding decision for one weight depends on the rounding of others, meaning that strictly local (per-weight) rounding cannot exploit cross-element error cancellation. This insight motivates adaptive rounding that assign each weight an optimized rounding direction to jointly minimize the overall quantization error.

Several prior works proposed adaptive rounding schemes (Hubara et al., 2020; Nagel et al., 2020; Kim et al., 2024; Lee et al., 2024; 2025) by introducing a learnable rounding matrix that determines, for each weight, whether to round up or down from the nearest quantization bin. While effective

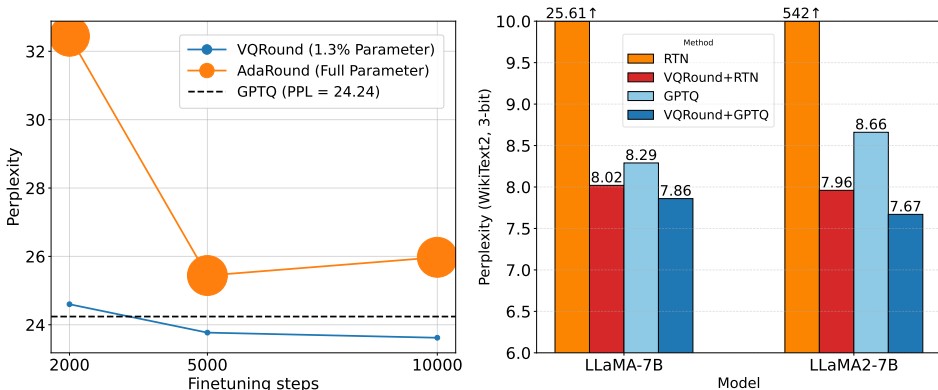

Figure 1: **Left:** Convergence behavior of Adaround and VQRound on OPT-350M under same steps; **Right:** WikiText2 perplexity of LLaMA-7B and LLaMA2-7B under 3-bit quantization.

on smaller networks (*e.g.,* 4-bit CNN quantization in AdaRound (Nagel et al., 2020), these methods scale poorly to LLMs because the rounding matrix is the same size as the weight matrix. This not only inflates memory and computation overhead, but also creates a huge solution space that is difficult to optimize (Shao et al., 2024; Ding et al., 2025). As a result, existing adaptive rounding approaches struggle to converge or yield marginal gains when applied to billion-parameter models. In practice, due to the difficulty of globally optimizing so many rounding parameters, many PTQ methods still resort to layer-wise or block-wise calibration (Nagel et al., 2020; Ding et al., 2025; Frantar et al., 2023), which breaks the problem into smaller subproblems at the cost of potentially losing global error cancellation benefits.

In this work, we address these limitations with VQRound, a new adaptive rounding framework tailored for large LLMs. The key idea is a vectorized reparameterization of the rounding matrix using vector quantization (VQ). Instead of assigning an independent learnable parameter to each weight, we represent the entire rounding matrix via a compact codebook. Each small group of weights is associated with a code from this codebook, whose entries encode rounding decisions. This design drastically shrinks the number of trainable parameters typically to around 0.2% of billion-parameter models, what naive per-weight rounding would require, thereby reducing the optimization complexity. Crucially, we find that proper initialization of the rounding codebook is vital for efficient convergence. By initializing the codebook such that the resulting quantized weights closely approximate the original full-precision weights, we minimize the initial quantization error and provide a strong starting point for subsequent fine-tuning. This careful initialization helps avoid bad local minima and accelerates the learning of rounding parameters. We demonstrate both theoretically and empirically that the proposed method is superior to other efficient reparameterization methods such as singular value decomposition (SVD) (Ding et al., 2025) and Kronecker product decomposition (Edalati et al., 2022).

Beyond the parameter reduction, VQRound introduces a lightweight end-to-end (E2E) fine-tuning stage to fully exploit cross-layer interactions. Instead of optimizing rounding in each layer independently, we jointly fine-tune all codebooks across the network using only a small calibration dataset (128 samples). In contrast to prior block-wise reconstructions, our E2E approach treats the quantized model's error holistically, yielding better accuracy with negligible data or compute overhead.

Moreover, VQRound is also designed as a plug-and-play component that can integrate with existing quantization workflows. Because it focuses solely on replacing the atomic rounding operation, it can complement orthogonal techniques such as group-wise quantization, outlier suppression (*e.g.,* weight clipping or rotation), and error compensation. Practically, one can insert VQRound into a standard PTQ pipeline (in place of RTN) to boost accuracy without modifying other parts of the algorithm. Our experiments on diverse LLMs (including, OPT (Zhang et al., 2022), LLaMA (Touvron et al., 2023a), LLaMA2 (Touvron et al., 2023b) and Qwen-3 (Yang et al., 2025)) show that VQRound consistently improves low-bit (4-bit, 3-bit, even 2-bit) quantization performance.

In summary, the main contributions of this paper are as follows:

- We propose VQRound, a novel vectorized reparameterization of the rounding matrix that drastically reduces the number of learned parameters (to around 0.2% of billion-parameter models)

while preserving fidelity. A careful initialization scheme is used to ensure the rounding codebook starts close to the full-precision model, which is critical for fast convergence.

- We theoretically analyze why the vectorized reparameterization method is superior to other efficient reparameterization methods, such as SVD and Kronecker product decomposition. Empirical results also support the theoretical analysis.
- We develop a lightweight end-to-end fine-tuning pipeline that globally optimizes all rounding codebooks using minimal calibration data.
- VQRound is designed as a modular drop-in replacement for standard rounding operations, making it compatible with and complementary to existing quantization frameworks.

## 2 RELATED WORK

### 2.1 ATOMIC ROUNDING

Rounding is the atomic operation in almost all LLM quantization algorithms (in contrast, vector quantization addresses a sphere packing problem via K-means clustering or nearest-neighbor search Egiazarian et al. (2024); Tseng et al. (2024). Thus, rounding is not needed). Delicate quantization methods typically employ round-to-nearest (RTN) as the atomic operation at different granularities, *i.e.,* per-tensor, per-group, per-channel, or even element-wise. Despite its central role, few studies investigate how to optimize rounding itself for efficient and accurate LLM quantization. Recent work shows that stochastic rounding (Croci et al., 2022) and adaptive rounding (Gupta et al., 2015) can effectively cancel out quantization error, yielding better solutions than RTN.

In particular, adaptive rounding replaces rigid nearest-neighbor rounding with learnable or optimized rounding functions that minimize task-relevant reconstruction loss. Early works such as AdaRound (Nagel et al., 2020) demonstrated that optimized rounding enables 4-bit quantization of CNNs, while subsequent methods like AdaQuant (Hubara et al., 2020) and BRECQ (Li et al., 2021) extended the framework with more flexible formulations, error metrics, and data-aware objectives. However, these do not show that adaptive rounding is ripe for scaling up in LLM quantization. Even though existing efforts such as (Lee et al., 2024) and (Lee et al., 2025) have made attempts to apply adaptive rounding in LLMs, the problems that will arise in scaling are unavoidable. For example, (Shao et al., 2024) mentioned that the rounding matrix is hard to optimize in LLM due to its huge solution space. What's more, the reconstruction is usually block-by-block (Wu et al., 2025), which would bring potentially high costs on large models.

### 2.2 VECTOR QUANTIZATION

Beyond model quantization, vector quantization has been widely adopted as a discrete representation learning mechanism in generative modeling. VQ-VAE (van den Oord et al., 2017) learns a discrete latent codebook that supports high-fidelity reconstruction and autoregressive priors, while VQ-GAN (Esser et al., 2021) couples a learned codebook in order to synthesize high-resolution images. In embodied AI, UniAct (Zheng et al., 2025) introduces a universal action space and action tokenizer that discretizes continuous robot controls into transferable action tokens.

VQ has emerged as a powerful alternative to scalar quantization for compressing large models. Unlike scalar methods that treat weights independently, VQ learns a codebook of representative vectors and maps groups of parameters to code indices. Recent methods adapt VQ to LLMs. AQLM (Egiazarian et al., 2024) performs input-adaptive multi-codebook quantization. VPTQ (Liu et al., 2024) introduces a second-order optimization framework. QuIP# (Tseng et al., 2024) combines Hessian-aware compression with task-aware reconstruction. While effective at very low bitwidths, these approaches typically require expensive Hessian estimation or multi-codebook clustering, making them computationally and memory intensive for PTQ. Different from the previous work, we use VQ to reparameterize the rounding matrix, which proves to be both efficient and accurate.

## 3 METHODOLOGY

In this section, we introduce the details of VQRound, which dramatically improves the efficiency of adaptive rounding. We utilize a vector codebook to reparameterize the rounding matrix as illustrated

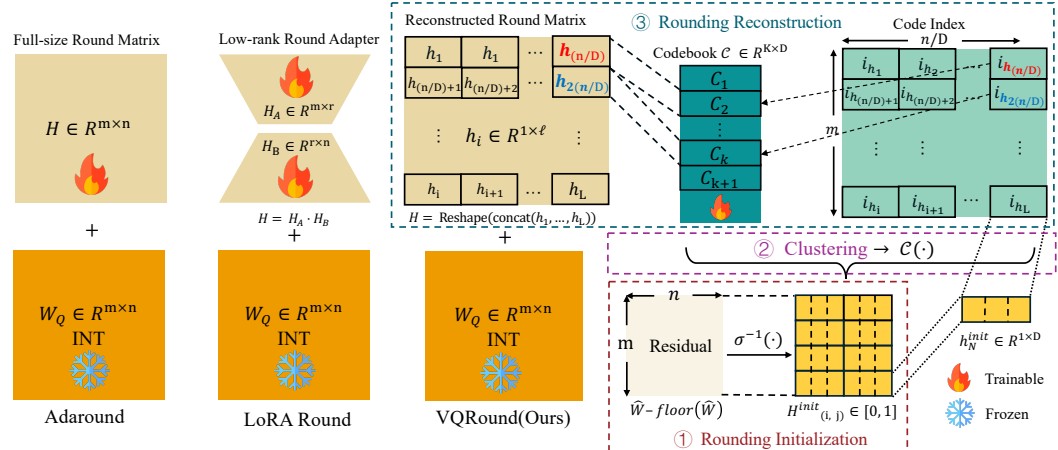

Figure 2: Comparison of different rounding strategies (Adaround (Nagel et al., 2020), LoRA Round and VQRound). Rounding matrix is initialized by the residual of quantized matrix $\hat{W}$ and its floored $floor(\hat{W})$ and $\sigma^{-1}(\cdot)$ is the inverse rectified sigmoid transform. In VQRound, only the codebook $\mathcal{C} \in \mathbb{R}^{N \times \ell}$ needs to be updated, more parameter-efficient than Adaround and LoRA Round.

in Fig. 2. In §3.2, we discuss the rationale for this approach and its optimality over other reparam-eterization methods. To cope with the suboptimal localized optimization of traditional adaptive rounding methods, we introduce an end-to-end finetuning approach that enables globally optimal quantization in a one-time finetuning stage in §3.4. In §3.5, we explain that VQRound can be used as a plug-and-play replacement for RTN.

**Notation.** We denote the weight matrix by $W$, its quantized form by $W_Q$ (or simply $Q$), the $j$-th column of $W$ by $W_j$, and the submatrix consisting of columns from the $j$-th onward (inclusive) by $W_{j:}$. Matrices are denoted by uppercase letters while scalars are represented by lowercase letters. The Hessian matrix is denoted by $\mathbf{H}$ throughout this paper.

## 3.1 PRELIMINARIES

We follow standard uniform affine weight quantization. For a weight matrix $W \in \mathbb{R}^{m \times n}$ and $b$-bit integers, the quantized integer tensor $W_Q$ under RTN is

$$W_Q = \text{clip}(\text{round}(\frac{W}{s}) + z, q_{min}, q_{max}), \tag{1}$$

where $s$ is the (per-tensor, -group or -channel) scale, $z$ is the zero-point, and $q_{\min}, q_{\max}$ denote the valid integer range. RTN independently rounds each entry of $W/s$ to its nearest integer.

**Adaptive rounding.** To enable error cancellation, adaptive rounding replaces the rigid RTN decision with a learnable rounding matrix $H \in [0, 1]^{m \times n}$ that controls the up/down rounding of each entry:

$$W_Q = \text{clip}\left(\left\lfloor \frac{W}{s} \right\rfloor + H + z, q_{\min}, q_{\max}\right), \qquad \widehat{W} = s\,(W_Q - z). \tag{2}$$

Here $H_{ij} = 0$ forces a round *down* (*e.g.*, $\lfloor 2.3 \rfloor = 2$), while $H_{ij} = 1$ rounds *up* (since $\lfloor 2.3 \rfloor + 1 = 3$). $H$ is initialized as the quantization residual, constrained to the interval $[0, 1]$, and gradually annealed toward binary values $0, 1$. At the inference stage, the resulting binary matrix provides deterministic rounding decisions. Classical adaptive rounding, however, learns $H$ as a dense matrix of the same size as $W$, which is impractical for LLMs.

**Element-wise Error.** In matrix approximation, the Frobenius norm $\|E\|_F = \left(\sum_{i,j} |E_{ij}|^2\right)^{1/2}$ measures the global energy of the error. For LLM quantization, however, such an energy-based metric can be misleading. The weight distributions in LLMs are often heavy-tailed, meaning that even if $\|E\|_F$ is small, a few large local errors can still occur and dominate the degradation in model performance. Let $H$ be the target (full-precision) rounding matrix, $\widehat{H}$ its approximation (quantized rounding matrix), and $E = H - \widehat{H}$ the error. We emphasize the element-wise worst-case error,

defined as $\|E\|_{\max} \triangleq \max_{i,j} |E_{ij}|$, and relate it to more global norms. For any matrix $E$, we have:

$$\|E\|_{\max} \leq \|E\|_2 \leq \|E\|_F, \tag{3}$$

where $\|\cdot\|2$ denotes the spectral norm. The first inequality follows from evaluating $x^\top E y$ on canonical basis vectors, and the second is the standard relationship between spectral and Frobenius norms (see Appendix A.4 for a proof). Thus, controlling $\|E\|_{\max}$ is at least as stringent as controlling $\|E\|_2$, and it also implies a bound on $\|E\|_F$. In practice, prioritizing a small $\|E\|_{\max}$ (*i.e.,* minimizing the worst-case per-weight error) curbs outlier rounding decisions and has been found to correlate better with downstream performance (e.g. lower perplexity in language modeling).

## 3.2 VECTORIZED REPARAMETERIZATION

Optimizing a full rounding matrix $H \in [0,1]^{m \times n}$ assigns one learnable variable to every weight and is therefore impractical at billion scale: the search space is $\mathcal{O}(mn)$ and the memory/compute footprint grows accordingly, making adaptive rounding pipelines slow or even infeasible on large LLMs. To address this, we propose a vectorized reparameterization of the rounding matrix that is far more parameter-efficient than AdaRound. In particular, we reparameterize $H$ using vector quantization (VQ), so that only a small codebook of vectors is learned.

We divide $H$ into $L$ vectors of length $D$, $\{h_\ell \in \mathbb{R}^D\}_{\ell=1}^L$ (so $L \cdot D = mn$), and learn a codebook $\mathcal{C} = \{c_1, \ldots, c_K\} \subset \mathbb{R}^D$ with $K \ll L$. Each vector is assigned to its nearest centroid

$$i_\ell = \arg \min_{k \in \{1,\ldots,K\}} \|h_\ell - c_k\|_2^2, \qquad h_\ell^{\text{VQ}} = c_{i_\ell},$$

yielding the VQ-based reconstruction of the rounding matrix $H_{\text{VQ}} = \text{reshape}\big([\,c_{i_1}, \ldots, c_{i_L}\,]\big)$. In forward passes, the substitution is a table lookup; only the codebook vectors $\{c_k\}$ are trainable. This reduces the number of learned parameters from $\mathcal{O}(mn)$ to $\mathcal{O}(KD)$ and the stored assignment to $L\lceil \log_2 K \rceil$ bits, while preserving local flexibility within each block.

Let $E_{\text{VQ}} = H - H_{\text{VQ}}$ be the quantization error after applying the codebook. Because each block $h_\ell$ is replaced by a single centroid $c_{i_\ell}$, the worst-case error over all weights is bounded by the largest error within any block: $\|E_{\text{VQ}}\|_\infty = \max_\ell \|h_\ell - c_{i_\ell}\|_\infty$. We can further prove that under mild regularity assumptions on the block distribution (see Appendix A.6), the minimum achievable worst-case error decays polynomially with the codebook size $K$ and exponentially with the block dimension $D$, *i.e.,*

$$\inf_{|\mathcal{C}|=K} \|E_{\text{VQ}}\|_\infty = \mathcal{O}\Big(K^{-1/D}\Big). \tag{4}$$

Larger codebooks (or smaller block sizes $D$) enable finer local adjustments, directly reducing the worst-case error $\|E\|_\infty$. In contrast, low-rank approximations such as SVD or Kronecker decomposition minimize the global energy metric $\|E\|_F$, which satisfy Eq. 3 but provide no guarantee on element-wise deviations. As a result, they may achieve small $\|E\|_F$ yet leave large local residuals, especially under heavy-tailed weight distributions. Our VQ-based reparameterization explicitly controls $\|E\|_\infty$, making it more robust to outliers. Empirically, we observe significantly smaller element-wise residuals after VQ initialization than with SVD or Kronecker decomposition (Fig. 4).

We can also prove a formal inequality to compare the best achievable worst-case error under our VQ approach versus low-rank methods (details in Appendix A.3 and A.7). In particular, we show that for a given budget of $N$ parameters, the minimum $\|\cdot\|_\infty$ error attainable with a codebook of size $N$ is asymptotically smaller than what any rank-constrained method can achieve:

$$\min_{|\mathcal{C}|=N} \|E_{VQ}\|_\infty \lesssim O(N^{-1/s}) < \min \|E_{LoRA/Kron}\|_\infty \leq \sigma_{r+1}; \quad \sigma_{r+1} \to \sigma_2(Kronecker) \tag{5}$$

by defining the LoRA and Kronecker rounding approximation as:

$$H_{\text{LoRA}} = AB^\top, \qquad A \in \mathbb{R}^{m \times r}, B \in \mathbb{R}^{n \times r}, \tag{6}$$

$$H_{\text{Kron}} = A \otimes B, \qquad A \in \mathbb{R}^{a \times c}, B \in \mathbb{R}^{b \times d}, \tag{7}$$

where $m = a \times b$, $n = c \times d$. Through the above approach, we have transformed the optimization of a rounding matrix with the same weight into the optimization of a vector codebook $\mathcal{C}$, thereby achieving a reduction in parameters. For codebook initialization-based clustering, we perform a K-means search on the codebook of each layer.

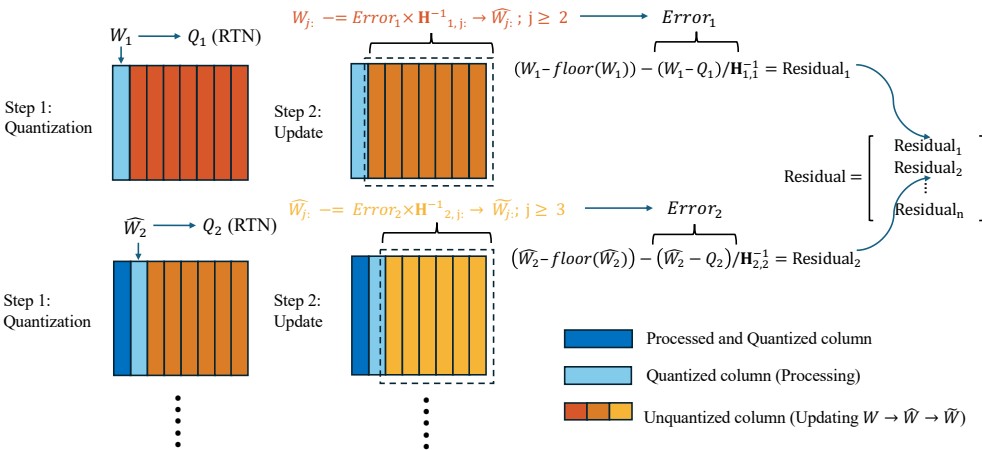

Figure 3: Initialization of rounding matrix based on residual before inverse rectified sigmoid.

### 3.3 ROUNDING INITIALIZATION

We find that the initialization of the rounding matrix is decisive for adaptive rounding: with identical training schedules, different initializations can lead to remarkably different final perplexities. Because rounding parameters determine whether each weight rounds up or down, a poor starting point can induce unstable optimization, slow convergence, and suboptimal rounding patterns that accumulate large quantization error. An effective initialization should (i) reduce inference error from the outset and (ii) explicitly control the worst-case element-wise deviation, *i.e.,* $\|E\|_\infty$ for $E = H - \widehat{H}$, so that each entry is already oriented toward a correct descent direction.

**Hessian-aware residual initialization.** We visualize our initialization scheme for channel-wise quantization with error compensation in Fig.3. At each step we (i) quantize the current column by RTN, (ii) form its residual, (iii) propagate a Hessian-scaled correction to the unprocessed columns, and (iv) convert the corrected residual to the initial rounding matrix through the inverse sigmoid as shown in Fig.2. This Hessian-aware initialization substantially reduces the initial deviation from the FP weights, stabilizes optimization, and yields better final quantization. Note that this initialization strategy naturally generalizes to group-wise Huang et al. (2024) or tensor-wise quantization by varying the group size. When the group spans the entire tensor, it reduces to a simple refinement of tensor-wise RTN.

### 3.4 END-TO-END FINETUNING

While VQ provides a compact initialization of the rounding matrix $H$, we observe that further E2E finetuning can directly refine the codebook $\mathcal{C}$, leading to better reconstruction of $H$ and improved quantization fidelity. Different from block-wise (Nagel et al., 2020; Wu et al., 2025) or layer-wise (Frantar et al., 2023; Tseng et al., 2024) reconstruction strategies, which optimize rounding decisions locally and are prone to sub-optimal minima, our E2E scheme jointly updates the rounding parameters across all layers under a global objective. This enables cross-layer error compensation and avoids local optima caused by independent block-level calibration.

Specifically, after the initialization of $H$ and reparameterization with VQ, we freeze the original weights and only update the codebook entries. During finetuning, a pretrained teacher model $\mathcal{M}_t$ is used to provide reference logits. Given an input batch $x$, the quantized student model $\mathcal{M}_s$ produces logits $\hat{y}$, while the teacher produces $y$. We minimize the following objective:

$$\mathcal{L} = \underbrace{\text{KL}\big(\text{softmax}(\hat{y}/T) \,\|\, \text{softmax}(y/T)\big)}_{\text{distillation loss}} + \lambda \cdot \underbrace{\mathcal{R}(H)}_{\text{rounding regularizer}}. \tag{8}$$

The distillation loss is computed by the KL-Divergence (Kullback & Leibler, 1951) between quantized student $\mathcal{M}_s$ and full-precision teacher model $\mathcal{M}_t$, where $T$ is the temperature for distillation, and $\mathcal{R}(H)$ encourages the relaxed rounding variables to approach hard $\{0, 1\}$ decisions (*e.g.,* $\mathcal{R}(H) = \sum_{i,j}[1 - |2H_{ij} - 1|^\beta]$). During E2E finetuning, $\beta$ is set as an annealing parameter which decay through steps. An excessively large beta constant can make the rounding loss difficult to

converge, while an excessively small rounding loss may result in an overly sharp rounding matrix, potentially leading to overfitting. The hyperparameter $\lambda$ balances task fidelity and rounding convergence. In Alg. 1, we show how do we implement VQRound's end-to-end finetuning process.

### 3.5 PLUG-AND-PLAY REPLACEMENT

VQRound serves not only as a PTQ technique for adaptive rounding, but also as a plug-and-play module that can be seamlessly incorporated into existing weight quantization frameworks. For instance, when applied to GPTQ, the process begins by running GPTQ on the full-precision model to estimate the layer-wise grid parameters, including channel-wise quantization factors(scale, zero point), updated GPTQ weights, and propagated residuals as described in §3.3. Each linear layer is then replaced with a VQ module whose forward pass reconstructs weights in GPTQ format while initializing with the precomputed residuals.

## 4 EXPERIMENTS

We evaluate VQRound on a diverse set of language model families, including OPT (Zhang et al., 2022), LLaMA (Touvron et al., 2023a), and LLaMA2 (Touvron et al., 2023b), which are widely adopted in both large language model applications and quantization research (Frantar et al., 2023; Egiazarian et al., 2024). To further examine its robustness, we additionally report results on the recent and more advanced Qwen3 model (Yang et al., 2025). We also validate the plug-and-play compatibility of VQRound by integrating it with existing quantization frameworks such as GPTQ (Frantar et al., 2023) and QuaRot (Ashkboos et al., 2024), where it consistently improves performance. Detailed experimental settings are provided in Appendix A.8.

### 4.1 RESULTS

We conduct evaluations on WikiText2 (Merity et al., 2017) and C4 (Raffel et al., 2020) using perplexity as the primary metric. In addition, zero-shot evaluations are performed on WinoGrande (Sakaguchi et al., 2019), PiQA (Bisk et al., 2020), HellaSwag (Zellers et al., 2019), and ARC-Easy/ARC-Challenge (Clark et al., 2018), with accuracy used as the evaluation criterion.

As shown in Tab. 1, VQRound has shown comparable performance with GPTQ on both 4-bit and 3-bit quantization, and it even outperforms GPTQ on OPT-125M, 350M, and 6.7B. When combined with GPTQ, VQRound generally improves quantization performance, outperforming GPTQ in most settings. These results validate the effectiveness of our method for low-bit quantization and highlight its plug-and-play compatibility with existing approaches. More results on the C4 dataset (Tab. 8) exhibit the same characteristics as WikiText2. Both results show that VQRound has good compensation on RTN and GPTQ. Experiments on QuaRot (Tab. 2) further validate the rationality of our plug-and-play design, confirming its seamless compatibility with existing quantization frameworks.

We further evaluate VQRound on the LLaMA and LLaMA2 families (Tab. 3). At 4-bit precision, it achieves results on par with GPTQ, while at 3-bit it consistently outperforms GPTQ (*e.g.,* reducing perplexity by 0.43 on LLaMA-7B and 0.99 on LLaMA2-7B). Under the extreme 2-bit setting where GPTQ collapses, VQRound remains stable with perplexity below 100, demonstrating its robustness and potential for ultra-low-bit quantization.

Table 1: OPT perplexity on Wikitext2. Lower is better.

| Precision | Method | OPT Model Size | | | | | |
|---|---|---|---|---|---|---|---|
| | | 125M | 350M | 1.3B | 2.7B | 6.7B | 13B |
| FP16 | Baseline | 27.65 | 22.00 | 14.63 | 12.47 | 10.86 | 10.13 |
| 4 bits | RTN | 37.29 | 25.94 | 48.17 | 16.92 | 12.10 | 11.32 |
| | VQRound+RTN | 30.69 | 23.77 | 15.48 | 13.30 | 11.26 | 10.66 |
| | GPTQ | 31.12 | 24.24 | 15.47 | 12.87 | 11.39 | **10.31** |
| | VQRound+GPTQ | **30.39** | **23.02** | **15.38** | **12.77** | **11.13** | 10.37 |
| 3 bits | RTN | 1.3e3 | 64.57 | 1.3e4 | 1.6e4 | 5.8e3 | 3.4e3 |
| | VQRound+RTN | 47.02 | 33.63 | 22.67 | 18.57 | 13.72 | 12.28 |
| | GPTQ | 53.85 | 33.79 | 20.97 | 16.88 | 14.86 | 11.61 |
| | VQRound+GPTQ | **46.10** | **28.03** | **19.13** | **15.55** | **12.45** | **11.37** |

Table 2: Our plug-and-play VQRound on QuaRot improves Wikitext2 perplexity under W4A16 asymmetric quantization (Ashkboos et al., 2024).

| Method | LLaMA | LLaMA2 |
|---|---|---|
| | 7B | 7B |
| FP16 | 5.68 | 5.47 |
| RTN | 7.94 | 6.99 |
| QuaRot+RTN | 7.46 | 6.76 |
| QuaRot+VQRound | **5.98** | **5.84** |

Table 3: LLaMA family perplexity on Wikitext2 and C4.

| Precision | Method | LLaMA-7B | | LLaMA-13B | | LLaMA2-7B | | LLaMA2-13B | |
| --- | --- | --- | --- | --- | --- | --- | --- | --- | --- |
| | | WikiText2 | C4 | WikiText2 | C4 | WikiText2 | C4 | WikiText2 | C4 |
| FP16 | Baseline | 5.68 | 7.34 | 5.09 | 6.80 | 5.47 | 7.26 | 4.88 | 6.73 |
| 4 bits | RTN | 6.29 | 8.12 | 5.53 | 7.23 | 6.12 | 8.17 | 5.21 | 7.14 |
| | VQRound+RTN | 6.13 | 7.88 | 5.42 | 7.17 | 5.90 | 7.88 | 5.19 | 7.13 |
| | GPTQ | 6.17 | 7.80 | **5.37** | 7.28 | 6.06 | 7.84 | **5.16** | **7.03** |
| | VQRound+GPTQ | **6.08** | **7.78** | 5.40 | **7.10** | **5.85** | **7.79** | 5.18 | 7.06 |
| 3 bits | RTN | 25.61 | 30.86 | 11.78 | 14.46 | 542.0 | 527.2 | 10.69 | 13.87 |
| | VQRound+RTN | 8.02 | 10.29 | 6.71 | 8.88 | 7.96 | 10.54 | 6.58 | 8.94 |
| | GPTQ | 8.29 | 10.51 | 6.73 | 8.83 | 8.66 | 11.24 | 6.55 | 8.76 |
| | VQRound+GPTQ | **7.86** | **9.95** | **6.46** | **8.47** | **7.67** | **10.05** | **6.33** | **8.57** |
| 2 bits | RTN | 1.1e5 | 1.1e5 | 5.7e4 | 5.9e4 | 1.8e4 | 5.1e4 | 2.8e4 | 5.3e4 |
| | VQRound+RTN | 65.41 | 43.52 | 47.57 | 31.53 | 84.07 | 56.67 | 68.27 | 38.54 |
| | GPTQ | 1.0e4 | 872.7 | 3.7e3 | 809.7 | 7.5e3 | 1.7e3 | 2.1e3 | 560.7 |
| | VQRound+GPTQ | **64.82** | **37.49** | **34.62** | **25.20** | **73.08** | **45.13** | **48.29** | **29.68** |

Table 4: Qwen3 perplexity on Wikitext2 and C4.

| Precision | Method | 0.6B | | 1.7B | | 4B | | 8B | |
| --- | --- | --- | --- | --- | --- | --- | --- | --- | --- |
| | | WikiText2 | C4 | WikiText2 | C4 | WikiText2 | C4 | WikiText2 | C4 |
| FP16 | Baseline | 20.96 | 30.31 | 16.67 | 22.36 | 13.64 | 19.83 | 9.72 | 15.42 |
| 4 bits | RTN | 37.39 | 51.69 | 28.26 | 32.45 | 17.47 | 24.57 | 12.01 | 18.48 |
| | VQRound+RTN | 25.55 | 35.30 | **16.97** | 25.08 | **13.57** | 21.59 | 10.33 | 16.76 |
| | GPTQ | 30.05 | 42.71 | 25.60 | 30.68 | 14.82 | **20.88** | 10.59 | 16.43 |
| | VQRound+GPTQ | **24.72** | **34.28** | 17.00 | **24.15** | 13.73 | 20.93 | **10.18** | **16.28** |

As shown in Tab. 4, VQRound demonstrates clear advantages on the Qwen3 family. It consistently mitigates the degradation of RTN and complements GPTQ, yielding stable improvements across model scales, with only negligible gaps in rare cases (*e.g.,* C4 on Qwen3-4B). This confirms that VQRound generalizes effectively to modern architectures.

We report zero-shot evaluation on five commonsense reasoning benchmarks in Tab. 5. As expected, quantization introduces some degradation, yet VQRound achieves performance largely comparable to GPTQ. Even for LLaMA2-13B, where the average gap to the full-precision model is the largest, the difference remains within 2%. These results confirm that VQRound preserves strong generalization ability across downstream reasoning tasks.

Table 5: 4 bit zero-shot accuracy (%) on commonsense benchmarks. Higher is better.

| Model | Method | WinoGrande ↑ | PiQA ↑ | HellaSwag ↑ | ArcE ↑ | ArcC ↑ | Average ↑ |
| --- | --- | --- | --- | --- | --- | --- | --- |
| LLaMA-7B | FP16 | 69.93 | 78.67 | 56.97 | 75.21 | 41.89 | 64.53 |
| | VQRound+RTN | **70.01** | 77.53 | 55.36 | 73.86 | 39.25 | 63.20 |
| | GPTQ | 69.93 | 77.86 | **55.99** | 74.12 | 39.51 | **63.48** |
| | VQRound+GPTQ | 68.59 | **78.18** | 55.17 | 73.44 | **40.10** | 63.10 |
| LLaMA-13B | FP16 | 72.77 | 79.16 | 59.92 | 77.40 | 46.42 | 67.13 |
| | VQRound+RTN | 71.59 | 78.51 | 58.45 | **76.52** | 45.05 | 66.02 |
| | GPTQ | 72.77 | **79.11** | **58.98** | 76.26 | 45.39 | **66.50** |
| | VQRound+GPTQ | **72.85** | 78.84 | 58.78 | 75.72 | **45.65** | 66.37 |
| LLaMA2-7B | FP16 | 69.06 | 78.07 | 57.13 | 76.30 | 43.43 | 64.80 |
| | VQRound+RTN | 68.11 | 76.88 | 55.55 | 73.36 | 40.27 | 62.83 |
| | GPTQ | **68.59** | 76.88 | **55.87** | **75.13** | **41.13** | **63.52** |
| | VQRound+GPTQ | 68.35 | **77.20** | 55.47 | 73.86 | 40.27 | 63.03 |
| LLaMA2-13B | FP16 | 72.38 | 79.05 | 60.07 | 79.38 | 48.46 | 67.87 |
| | VQRound+RTN | **72.22** | **78.94** | **59.21** | 77.65 | **45.90** | **66.78** |
| | GPTQ | 70.96 | 78.02 | 58.74 | 77.44 | **45.90** | 66.21 |
| | VQRound+GPTQ | 72.14 | 78.73 | 59.14 | **78.11** | 45.39 | 66.70 |

Table 6: Initialization Comparison on Perplexity. (a) Residual initialization comparison on different weight integer calculation. (b) Rounding strategy comparison across LoRA Round with Kaiming or SVD initialization, Kronecker product, and Vector codebook. For LoRA, we use rank=64.

(a) Residual initialization.

| Init Method | OPT Model Size | | | |
| | 125M | | 350M | |
| | Soft | Hard | Soft | Hard |
|---|---|---|---|---|
| $W/s$ | 28.54 | 38.58 | 22.63 | 28.42 |
| $W_Q/s$ | 63.54 | 58.04 | 23.95 | 27.92 |
| $W_Q/s$ w. $\mathbf{H}$ | 46.11 | 40.85 | 23.42 | 24.36 |

(b) Rounding matrix initialization on OPT-125M.

| Method | Init Eval | |
| | Soft | Hard |
|---|---|---|
| LoRA (Kaiming) | 90.82 | 5665.40 |
| LoRA (SVD) | 128.11 | 35.08 |
| Kronecker | 156.47 | 5665.40 |
| VQ | 28.54 | 38.58 |

Table 7: (a) Trainable parameters comparison between AdaRound and VQRound. OOM means out of memory, and NAN means no result. (b) VQRound 4-bit results for different codebook settings on OPT-1.3B. K stands for the total number of codebook centroids, and D is the length of each vector.

(a) Trainable parameters comparison.

| Model | Trainable Params | | VQ/Ada |
| | AdaRound | VQRound | Ratio (%) |
|---|---|---|---|
| OPT-1.3B | 1.21B | 4.72M | 0.39% |
| OPT-2.7B | 2.16B | 6.29M | 0.29% |
| LLaMA-7B | 6.48B | 7.34M | 0.11% |
| LLaMA-13B | OOM | 9.18M | NAN |

(b) Codebook setting comparison.

| Codebook Setting | PPL | |
| | WikiText2 | C4 |
|---|---|---|
| $K = 2^{12}, D = 4$ | 15.84 | **17.10** |
| $K = 2^{12}, D = 8$ | **15.48** | 17.28 |
| $K = 2^{16}, D = 4$ | 15.73 | 17.11 |
| $K = 2^{16}, D = 8$ | 16.13 | 17.23 |

## 4.2 ABLATION STUDY

We investigate the impact of different initialization strategies on model performance. As discussed in §3.3 and §3.2, initialization plays a critical role in effective optimization. In our ablations, we examine alternative designs for both the residual and the rounding matrix (Fig. 2). For residual representation, we consider two factors: whether to incorporate the original full-precision weights $W_{FP}$ and whether to leverage Hessian information $\mathbf{H}$. The results in Tab. 6a show that Hessian-informed initialization consistently achieves lower reconstruction error than methods without Hessian guidance. While including $W_{FP}$ further reduces the initial error, it does not improve downstream optimization and may even hinder convergence. Finally, as reported in Tab. 6b, VQ reparameterization substantially outperforms LoRA Round and Kronecker-based initialization, underscoring its effectiveness as a reparameterization method.

We investigate the trade-off between the codebook size $K$ and the vector dimension $D$. As shown in Tab. 7, the configuration with $K = 2^{12}, D = 8$ achieves the lowest perplexity on WikiText2. In contrast, using a smaller $D$ increases the memory footprint, while a larger $K$ leads to more trainable parameters, making $K = 2^{12}, D = 8$ a balanced choice between efficiency and performance.

## 5 CONCLUSION

In this work, we present VQRound, a vectorized reparameterization method for adaptive rounding in post-training quantization. By reducing the number of learnable parameters to less than 0.2% of the billion-parameter models while maintaining performance, VQRound achieves both efficiency and effectiveness. A dedicated initialization strategy further stabilizes training by aligning the codebook with the full-precision model, which is also critical for fast convergence. We provide theoretical and empirical evidence demonstrating the superiority of vectorized reparameterization over alternatives such as SVD and Kronecker decompositions. In addition, we introduce a lightweight fine-tuning pipeline that globally optimizes codebooks with limited calibration data, making the approach highly practical. Finally, VQRound is designed as a modular plug-and-play component, ensuring compatibility with existing quantization frameworks and enabling integration with methods such as GPTQ and QuaRot. Overall, VQRound advances the design of efficient rounding mechanisms for low-bit quantization, offering a principled and versatile solution that combines theoretical rigor, empirical performance, and practical usability.

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

## A  APPENDIX

### A.1  ADDITIONAL EXPERIMENT RESULT

Table 8: C4 Perplexity in OPT model family. Lower is better.

| Precision | Method | OPT Model Size | | | | | |
|---|---|---|---|---|---|---|---|
| | | 125M | 350M | 1.3B | 2.7B | 6.7B | 13B |
| FP16 | Baseline | 26.56 | 22.59 | 16.07 | 14.34 | 12.71 | 12.06 |
| 4 bits | RTN | 33.91 | 26.21 | 24.51 | 18.43 | 14.36 | 13.36 |
| | VQRound+RTN | 28.79 | 24.39 | 17.28 | 15.27 | 13.27 | 12.53 |
| | GPTQ | 29.22 | 24.63 | 16.97 | 15.00 | 13.18 | **12.26** |
| | VQRound+GPTQ | **28.72** | **23.44** | **16.80** | **14.87** | **13.01** | 12.29 |
| 3 bits | RTN | 834 | 55.49 | 5.2e3 | 1.1e4 | 5.3e3 | 3.1e3 |
| | VQRound+RTN | 39.76 | 31.40 | 22.57 | 19.28 | 15.57 | 14.37 |
| | GPTQ | 42.41 | 31.33 | 21.63 | 18.17 | 17.14 | 13.34 |
| | VQRound+GPTQ | **38.87** | **27.13** | **20.02** | **17.16** | **14.25** | **13.18** |

### A.2  ALGORITHM OF VQROUND END-TO-END (E2E) FINETUNING

---
**Algorithm 1** VQRound end-to-end finetuning

---
**Require:** Teacher model $\mathcal{M}_t$, Student model $\mathcal{M}_s$; Frozen FP weights $\mathbf{W}$, per-channel scale $S$, zero-point $Z$; fixed VQ indices $I = \{i_\ell\}_{\ell=1}^{L}$; initial codebook $\mathcal{C} = \{c_k\}_{k=1}^{K}$; calibration dataset $\mathcal{D}$; temperature $T$; rounding regularizer $\mathcal{R}(\cdot)$; regularization weight $\lambda$; steps $N$; Anneal parameter $(\beta_{\text{high}}, \beta_{\text{low}})$

   **Freeze** all params of $\mathcal{M}_s$ except the codebook
   **Init Adam optimizer** on $\mathcal{C}$
   **for** $t \leftarrow 1$ **to** $N$ **do**
      $x \leftarrow \text{NextSample}(\mathcal{D})$; Batch size = 1
      $\hat{H} \leftarrow \text{Reshape}([\, c_{i_1}, \ldots, c_{i_l} \,])$
      $\hat{W} \leftarrow \big( \text{clip}(\lfloor W/S \rfloor + \hat{H} + Z,\, 0,\, Q_{\max} - Z) \big)$
      $\hat{y} \leftarrow \mathcal{M}_s(x; \hat{W}); \quad y \leftarrow \mathcal{M}_t(x)$
      $\mathcal{L}_{\text{KD}} \leftarrow \text{KL}\big(\text{softmax}(\hat{y}/T) \,\|\, \text{softmax}(y/T)\big)$
      $\beta_t \leftarrow \text{Anneal}\big(\beta_{\text{hi}}, \beta_{\text{lo}}, t, N\big)$
      $\mathcal{L} \leftarrow \mathcal{L}_{\text{KD}} + \lambda \, \mathcal{R}(\hat{H}; \beta_t)$
      **Update** $\mathcal{C} \leftarrow \text{Adam}(\mathcal{C}, \nabla_{\mathcal{C}})$
   **end for**

---

### A.3  EXPLANATION OF LORA ROUND WITH SVD INITIALIZATION AND KRONECKER ROUNDING

**Low-rank Adaptation (LoRA):** When we consider parameter-efficient reparameterization methods such as low-rank adaptation (LoRA), the rounding matrix $H$ is approximated via a multiplicative form:

$$H_{\text{LoRA}} = AB^\top, \qquad A \in \mathbb{R}^{m \times r}, \; B \in \mathbb{R}^{n \times r}. \tag{9}$$

Denote the approximation error by $E_{LoRA} := H - H_{LoRA}$. We can initiate the low-rank approximation of rounding matrix through SVD decomposition by defining $H = \sum_{k \geqslant 1} \sigma_k u_k v_k^\top$ with $\sigma_1 \geq \sigma_2 \geq \ldots$ and $H_{LoRA} = H_r = \sum_{k=1}^{r} \sigma_k u_k v_k^\top$. By the Eckart-Young-Mirsky theorem (Eckart & Young, 1936), we have (Detailed proof in Appendix A.5):

$$\|E_{LoRA}\|_F = \|H - H_r\|_F = \left( \sum_{k>r} \sigma_k^2 \right)^{1/2}. \tag{10}$$

also:

$$\|E_{LoRA}\|_2 = \|H - H_r\|_2 = \sigma_{r+1} \tag{11}$$

Hence according to Eq. 3, we have:

$$\|E_{LoRA}\|_\infty \leq \sigma_{r+1} \tag{12}$$

We can absorb that the SVD method can optimally minimize the global error through the fixed Frobenius norm promised by Eckart–Young–Mirsky theorem. Yet, its infinity norm is constrained by the $(r+1)$th largest singular value. In most LLMs, due to the widespread presence of outliers in weights, even when decomposing the matrix at higher ranks as shown in Fig. 5, the singular values remain highly significant in relatively high rank (*e.g.,* $r = 64$). This prevents the element-wise error during quantization from being confined within a low range.

**Kronecker Product:** Kronecker product can be used to approximate rounding matrix $H$ by two smaller matrices $A \in \mathbb{R}^{a \times c}$ and $B \in \mathbb{R}^{b \times d}$ where $m = a \cdot b, n = c \cdot d$, the rounding matrix can be represented by:

$$H_{kron} = A \otimes B \tag{13}$$

Kronecker product is a relatively complex matrix operation, thus we denote its result through Van Loan-Pitsiantis rearrangement (Van Loan & Pitsianis, 1993): $R(A \otimes B) = vec(A)vec(B)^\top$. By doing this, we can still optimize the Kronecker product according to SVD: $R(H) = \sum_{k=1}^{r} \sigma_k u_k v_k^\top$. Due to the properties of this rearrangement:

$$\|H\|_F = \|R(H)\|_F \tag{14}$$

Based on the conclusion from Eq. 10, we found that its Frobenius norm satisfies:

$$\|E_{Kron}\|_\infty \leq \|E_{Kron}\|_F = \|H - H_{Kron}\|_F = (\sum_{k \geq 2} \sigma_k (R(H))^2)^{1/2} \tag{15}$$

Similarly, in the Kronecker product form, we can still only minimize the global error, its element-wise error is not guaranteed.

A.4    PROOF OF EQUATION 3

*Proof.* **Part 1: Proof of** $\|E\|_\infty \leq \|E\|_2$

First, we recall the definitions of the two norms. The element-wise infinity norm is the maximum absolute value of any element in the matrix:

$$\|E\|_\infty \triangleq \max_{i,j} |E_{ij}|$$

Let this maximum value be achieved at the entry $E_{rc}$, such that $\|E\|_\infty = |E_{rc}|$.

The spectral norm is defined as:

$$\|E\|_2 \triangleq \max_{\|\mathbf{x}\|_2=1} \|E\mathbf{x}\|_2$$

By its definition, for any vector $\mathbf{v}$ with $\|\mathbf{v}\|_2 = 1$, the inequality $\|E\mathbf{v}\|_2 \leq \|E\|_2$ must hold.

Let us choose a specific unit vector. Let $\mathbf{e}_c \in \mathbb{R}^n$ be the standard basis vector with a 1 in the $c$-th position and zeros elsewhere. Clearly, $\|\mathbf{e}_c\|_2 = 1$.

The product $E\mathbf{e}_c$ results in the $c$-th column of the matrix $E$. Let's denote this column vector as $\mathbf{col}_c(E)$.

$$E\mathbf{e}_c = \begin{pmatrix} E_{1c} \\ E_{2c} \\ \vdots \\ E_{mc} \end{pmatrix}$$

Applying the definition of the spectral norm, we have:

$$\|E\|_2^2 \geq \|E\mathbf{e}_c\|_2^2$$

The squared Spectral-norm of the $c$-th column is the sum of the squared absolute values of its elements:

$$||E\mathbf{e}_c||_2^2 = \sum_{i=1}^{m} |E_{ic}|^2$$

This sum includes the term $|E_{rc}|^2$, which is the square of the infinity norm. Since all terms in the sum are non-negative:

$$\sum_{i=1}^{m} |E_{ic}|^2 \geq |E_{rc}|^2 = (||E||_\infty)^2$$

Combining these inequalities, we get:

$$||E||_2^2 \geq ||E\mathbf{e}_c||_2^2 \geq (||E||_\infty)^2$$

Taking the square root of both sides yields the desired result:

$$||E||_2 \geq ||E||_\infty$$

**Part 2: Proof of $||E||_2 \leq ||E||_F$**

We start with the definitions. The spectral norm is the largest singular value of $E$:

$$||E||_2 = \sigma_1(E)$$

The Frobenius norm is defined as the square root of the sum of the squared absolute values of all elements:

$$||E||_F = \sqrt{\sum_{i=1}^{m} \sum_{j=1}^{n} |E_{ij}|^2}$$

A fundamental property of matrix norms is that the squared Frobenius norm of a matrix is equal to the sum of its squared singular values. This arises from the fact that $||E||_F^2 = \mathrm{Tr}(E^H E)$, and the trace of $E^H E$ is the sum of its eigenvalues, which are the squared singular values of $E$.

$$||E||_F^2 = \sum_{k=1}^{\mathrm{rank}(E)} \sigma_k(E)^2$$

Let's expand this sum:

$$||E||_F^2 = \sigma_1(E)^2 + \sigma_2(E)^2 + \cdots + \sigma_{\mathrm{rank}(E)}(E)^2$$

Substituting the definition of the spectral norm, $||E||_2 = \sigma_1(E)$:

$$||E||_F^2 = ||E||_2^2 + \sum_{k=2}^{\mathrm{rank}(E)} \sigma_k(E)^2$$

Since all singular values are non-negative, their squares are also non-negative. Therefore, the sum of the remaining squared singular values must be non-negative:

$$\sum_{k=2}^{\mathrm{rank}(E)} \sigma_k(E)^2 \geq 0$$

This implies:

$$||E||_F^2 \geq ||E||_2^2$$

Taking the square root of both sides gives the second part of our inequality:

$$||E||_F \geq ||E||_2$$

Combining the results from Part 1 and Part 2, we have proven the complete inequality chain. □

## A.5    PROOF OF EQUATION 10 AND EQUATION 12

*Proof.* We begin by defining the error matrix $E_r = A - A_r$.

$$E_r = \left( \sum_{k=1} \sigma_k \mathbf{u}_k \mathbf{v}_k^T \right) - \left( \sum_{k=1}^{r} \sigma_k \mathbf{u}_k \mathbf{v}_k^T \right)$$
$$= \sum_{k=r+1} \sigma_k \mathbf{u}_k \mathbf{v}_k^T$$
$$= \sigma_{r+1} \mathbf{u}_{r+1} \mathbf{v}_{r+1}^T + \sigma_{r+2} \mathbf{u}_{r+2} \mathbf{v}_{r+2}^T + \dots$$

This expression for the error matrix is the foundation for both proofs.

### 1. PROOF OF THE SPECTRAL NORM ERROR

The spectral norm of a matrix, $||M||_2$, is defined as its largest singular value, $\sigma_1(M)$. Our goal is to find the singular values of the error matrix $E_r$.

The expression derived above, $E_r = \sum_{k=r+1} \sigma_k \mathbf{u}_k \mathbf{v}_k^T$. the singular values of the matrix $E_r$ are precisely the set $\{\sigma_{r+1}, \sigma_{r+2}, \dots, \sigma_N\}$. Thus, the largest singular value in this set is $\sigma_{r+1}$. By the definition of the spectral norm:

$$||E||_2 = \sigma_{r+1}$$

This completes the first part of the proof.

### 2. PROOF OF THE FROBENIUS NORM ERROR

For our error matrix $E_r = A - A_r$, from Part 1, we have already identified the singular values of $E_r$ to be $\{\sigma_{r+1}, \sigma_{r+2}, \dots, \sigma_N\}$. According to (Horn & Johnson, 2012):

$$||A - A_r||_F^2 = \sum_{k=r+1}^{N} \sigma_k^2$$

Taking the square root of both sides, we obtain the desired result:

$$||A - A_r||_F = \left( \sum_{k=r+1}^{N} \sigma_k^2 \right)^{1/2}$$

This completes the second part of the proof. $\qquad\square$

## A.6    PROOF OF EQUATION 4

*Proof.* The proof is established by deriving matching upper ($O$) bounds on the infimal covering radius, $r_\infty(C)$.

**1. Upper Bound** ($O(N^{-1/s})$)    We establish the upper bound via a constructive proof. We design a specific codebook $C'$ and show its covering radius is $O(N^{-1/s})$. The optimal codebook's performance must be at least as good.

Let the set $\Omega$ have a finite $s$-dimensional volume $V = \int_\Omega dx$. We can tile this volume with $N$ identical, non-overlapping $s$-dimensional hypercubes, $\{\mathcal{S}_k\}_{k=1}^{N}$. The volume of each hypercube is $V_k = V/N$.

Let $L$ be the side length of these hypercubes. The volume of an $s$-dimensional hypercube is $L^s$. Thus:

$$L^s = \frac{V}{N} \implies L = \left( \frac{V}{N} \right)^{1/s} = V^{1/s} N^{-1/s} \tag{16}$$

As $V$ is a constant, the side length scales as $L = O(N^{-1/s})$.

We construct our codebook $C'$ by placing one codeword $c_k'$ at the center of each hypercube $\mathcal{S}_k$. For any vector $x \in \mathcal{S}_k$, the quantizer maps it to $c_k'$. The $\ell_\infty$ error is the maximum coordinate-wise distance from $x$ to the center. For a hypercube of side length $L$, this maximum distance is $L/2$.

The covering radius of this constructed codebook is therefore:

$$r_\infty(C') = \sup_{x \in \Omega} \min_{c \in C'} ||x - c||_\infty = \max_k \sup_{x \in \mathcal{S}_k} ||x - c_k'||_\infty = \frac{L}{2} = O(N^{-1/s})$$

Since the infimum is the greatest lower bound, it must be less than or equal to the error of this particular construction:

$$\inf_{|C|=N} r_\infty(C) \le r_\infty(C') = O(N^{-1/s})$$

**Conclusion**    From the upper bound, we have $\inf r_\infty(C) \le O(N^{-1/s})$.

$$\inf_{|C|=N} r_\infty(C) = O(N^{-1/s})$$

Thus, $\inf_{|C|=N} ||E_{VQ}||_\infty = O(N^{-1/s})$ since $r_\infty(C) = \sup_{x \in \Omega} \min_{c \in C'} ||x - c||_\infty = \sup_{x \in \Omega} \min_{c \in C'} ||E_{VQ}||_\infty$ This completes the proof based on the given assumptions.

$\square$

### A.7    PROOF OF EQUATION 5

**Problem Setup**    Let the target matrix $H \in \mathbb{R}^{m \times n}$ be defined as $H = L + S$, where:

- $L$ is a matrix with $\text{rank}(L) = r$.
- $S = M \cdot \mathbf{e}_i \mathbf{e}_j^T$ is a matrix with a single non-zero entry $M$ at position $(i, j)$, where $M \gg ||L||_2$.

We analyze the minimal Frobenius and infinity-norm error for three approximation methods: LoRA ($H_{LoRA}$), Kronecker ($H_{Kron}$), and VQ ($H_{VQ}$).

**Lemma 1** (Error Bound for LoRA). *The minimal Frobenius error for the best rank-$r$ approximation of $H$ is lower-bounded by the magnitude of the outlier:*

$$\min_{rank(\hat{H}) \le r} ||H - H_{LoRA}||_F = \Omega(M)$$

*Proof.* We have (Horn & Johnson, 2012):

$$||E_{LoRA}||_F = ||(L - H_{LoRA}) + S||_F = ||L - H||_F + ||S||_F + 2\langle L + H, S \rangle$$

Since $S$ only has non-zero entry $M$ on $(i, j)$:

$$||E_{LoRA}||_F = ||L - H||_F + ||S||_F + 2(L - H)_{ij} \cdot M$$

Also:

$$||S||_F^2 = |S_{11}|^2 + |S_{12}|^2 + \cdots + |S_{ij}|^2 + \cdots = 0^2 + 0^2 + \cdots + |M|^2 + \cdots = |M|^2$$

$$||L - H||_F \simeq 0, \quad L - H \simeq 0$$

because both $L$ and $H$ has rank $r$. Thus, the error $||E_{LoRA}||_F \simeq 0 + M + 2(L - H)_{ij} \cdot M \simeq M$ In general, the error must be dominated by the magnitude of the outlier, which implies the minimal error is on the order of $M$ $\square$

**Lemma 2** (Error Bound for Kronecker Product). *Let $p = rank(R(L))$, where $R(\cdot)$ is the Van Loan-Pitsianis rearrangement. The minimal Frobenius-norm error for the best $p$-term Kronecker approximation of $H$ is lower-bounded by the magnitude of the outlier:*

$$\min_{p\text{-term}} ||H - H_{Kron}||_F = \Omega(M)$$

*Proof.* The proof follows the same logic as Lemma 1, but in the rearranged space. A Kronecker approximation $H_{Kron} = A \otimes B$ is equivalent to a rank-1 approximation of the rearranged matrix $R(H)$. The best such approximation is given by the truncated SVD of $R(H)$.

Let $\hat{R}$ be the best rank-1 approximation of $R(H)$. The minimal squared Frobenius-norm error is given by the Eckart-Young-Mirsky theorem:

$$\min_{p\text{-term}} ||H - H_{Kron}||_F^2 = \min ||R(H) - \hat{R}||_F^2 = \sum_{k \geq 2} \sigma_k(R(H))^2 \tag{17}$$

The sum of non-negative terms is always greater than or equal to its largest term. In this case, the largest term in the sum is the first one, corresponding to $k = 2$.

$$\sum_{k \geq 2} \sigma_k(R(H))^2 \geq \sigma_2(R(H))^2 \tag{18}$$

Therefore, we only need to show that the (2)-th singular value of $R(H)$ is lower-bounded by the magnitude of the outlier.

According to Van Loan-Pitsianis rearrangement's property (Van Loan & Pitsianis, 1993), we have:

$$R(H) = R(L + S) = R(L) + R(S)$$

Here, we have $rank(R(L)) = 1$. $R(S)$ is the rearrangement of a single-entry matrix, which is also a single-entry matrix of the form $M \cdot \mathbf{e}_{i'} \mathbf{e}_{j'}^T$. Thus, $R(S)$ is also a rank-1 matrix with spectral norm $||R(S)||_2 = M$. Given that $R(L)$ is a rank-1 matrix ($\sigma_2(R(L)) = 0$) and the perturbation $R(S)$ has a large norm ($||R(S)||_2 = M$), the 2-th singular value of their sum is lower-bounded by a significant value related to the perturbation. For an incoherent perturbation, it can be shown that:

$$\sigma_2(R(H)) = \sigma_2(R(L) + R(S)) = \Omega(M) \tag{19}$$

$\square$

**Lemma 3** (Error Bound for VQ). *The minimal infinity-norm error for a VQ approximation of $H$ with a codebook of size $N$ has an upper bound that is independent of the outlier magnitude $M$:*

$$\min_{|C|=N} ||H - H_{VQs}||_\infty = O(N^{-1/s})$$

*where $s$ is the dimension of the VQ blocks.*

This has been proved in Appendix A.6

**Theorem 1** (Comparative Analysis). *For a matrix with a large sparse outlier as defined, VQ offers an asymptotically superior approximation in the infinity norm compared to LoRA and Kronecker Product approximation.*

$$\min_{VQ} ||E||_\infty < \min_{Kron} ||E||_F \approx \min_{LoRA} ||E||_F$$

*Proof.* From Lemma 1 and Lemma 2, the minimal error for both LoRA and Kronecker is lower-bounded by a large constant of order $\Omega(M)$. From Lemma 3, the minimal error for VQ is upper-bounded by a term $O(N^{-1/s})$, which is independent of $M$ and decreases as the codebook size $N$ increases.

For a sufficient outlier magnitude $M$ ($M > 1$) and a reasonable codebook size $N$ (in out experiment, we set $N = 2^{12}, S = 8$), we have:

$$O(N^{-1/s}) < \Omega(M)$$

The theorem follows directly by combining these bounds. VQ's local adaptivity allows it to isolate the outlier, while the global nature of LoRA and Kronecker approximation leads to their failure. $\square$

## A.8   Experiment setting

To facilitate reproducibility, we detail the experimental settings and hyperparameters. For VQ initialization, a codebook of 4096 ($2^{12}$) centroids with vector dimension $D = 8$ is employed. Each layer undergoes 100 iterations of K-Means clustering, providing a balance between search quality and initialization efficiency. During end-to-end fine-tuning, the codebook is optimized with the Adam optimizer (Kingma & Ba, 2014). A unified hyperparameter configuration is adopted across models: the learning rate is set to $1e - 2$, the rounding regularization coefficient $\lambda$ to $1e - 2$, and the annealing schedule for $\beta$ decreases from 20 to 2. Fine-tuning is conducted for 5000 steps, with the first 10% used as a distillation-only warm-up phase to ensure stable convergence, after which the rounding loss is incorporated into the training objective. All experiments are performed on 128 randomly sampled sequences from the C4 dataset (Raffel et al., 2020) with length 2048. To accelerate initialization, GPU-accelerated K-Means clustering is implemented using FAISS (Douze et al., 2025). All experiments are conducted on a single NVIDIA RTX A6000 GPU.

## A.9   Figure

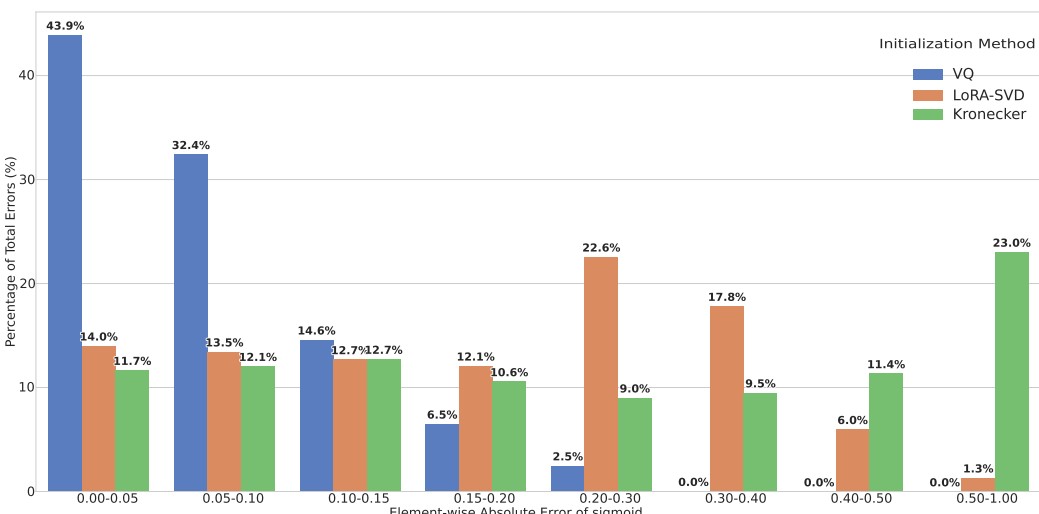

Figure 4: Quantization Initialization Error between different reparameterization methods. The error is computed by $\text{sigmoid}(H_{init}) - \text{sigmoid}(H_{reconstruct})$ so that the error is in [0,1]. Higher percentage of the low sigmoid error interval reveals lower initialization error.

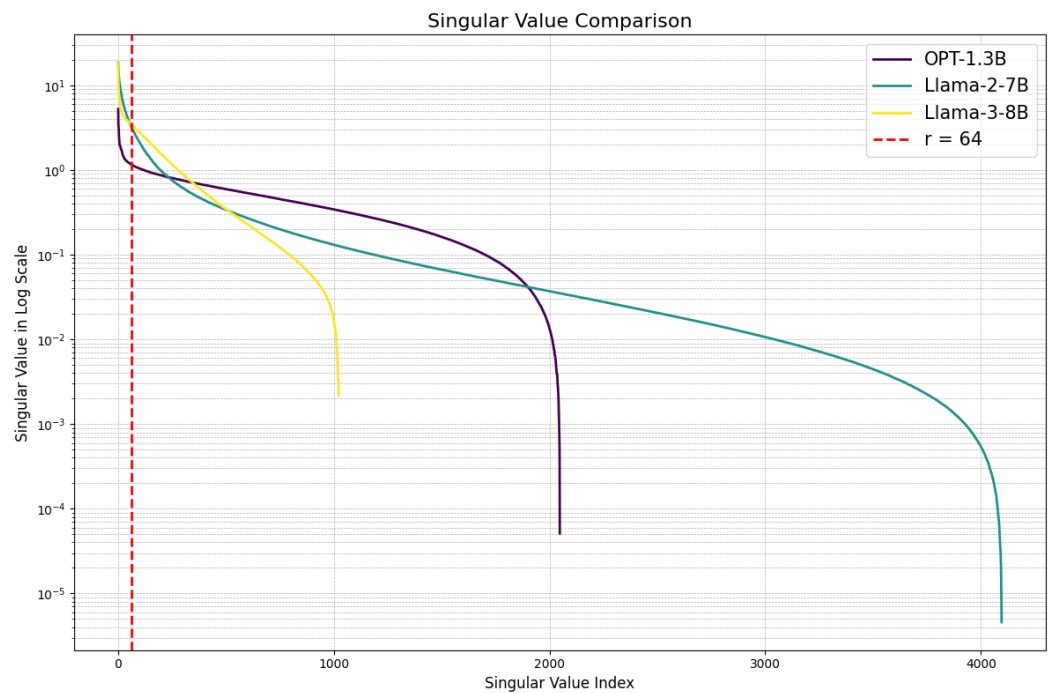

Figure 5: Singular value distribution

### A.10 USE OF LLM

This paper was written with the assistance of the LLM. There are some sentences or paragraphs revised with the support of the LLM. Whereas they have been fully reviewed and verified by the author. The authors accept the academic responsibility for the content of this work, including the parts where the LLM provided assistance.

