# OpenReview forum: "Revisiting Atomic Rounding with Vectorized Reparameterization for LLM Quantization"
_ICLR.cc/2026/Conference — ICLR 2026 Conference Withdrawn Submission_

### Official Review · Reviewer_EE7v · 2025-10-21

**Soundness:** 2
**Presentation:** 3
**Contribution:** 2
**Rating:** 4
**Confidence:** 5

**Summary:**

The paper proposes VQRound, a memory-efficient method for adaptive rounding in post-training quantization (PTQ) of large language models (LLMs). To address the prohibitive memory and optimization cost of full-size learnable rounding matrices (as in AdaRound), the authors reparameterize the rounding matrix using vector quantization (VQ): they cluster small blocks of rounding decisions into a compact codebook, drastically reducing trainable parameters (to ~0.2% of the original). They further introduce a Hessian-aware initialization scheme and a lightweight end-to-end fine-tuning pipeline using small amounts of samples.

**Strengths:**

1) The core idea of using VQ to compress the rounding matrix is well-motivated and yields significant parameter savings (e.g., 7.34M vs. 6.48B for LLaMA-7B), making adaptive rounding feasible for billion-scale models.

2) The method demonstrates non-trivial gains over strong baselines like GPTQ and QuaRot across multiple model families and bit-widths, including the challenging 2-bit regime where prior methods collapse.

**Weaknesses:**

1) The novelty of the work is limited. The paper combines two well-established ideas—adaptive rounding (AdaRound, BRECQ) and vector quantization (VQ-VAE, AQLM)—without a fundamentally new insight. While the application to rounding matrices is novel, the technical contribution is largely engineering: replacing a dense matrix with a VQ codebook. The theoretical analysis (e.g., ∥E∥∞ vs. ∥E∥F) is sound but standard in approximation theory and does not constitute a conceptual leap.

2) The end-to-end fine-tuning of large language models appears to partially undermine the memory efficiency advantage of VQRound, as it requires maintaining and backpropagating through the full model—even though only the small codebook parameters are updated.

3) The individual contribution of the end-to-end fine-tuning strategy (as opposed to local, layer-wise optimization used in methods like GPTQ) and the role of knowledge distillation in improving quantization accuracy remain unclear; a more thorough ablation study is needed to disentangle their effects.

**Questions:**

1) How about the results of the proposed method on reasoning models and multi-modal large models?

2) I guess the work is not as plug-and-play as claimed, particularly when integrated with other quantization frameworks that already incorporate fine-tuning—such as OSTQuant[1] and FlatQuant[2].

3) Can you go deeper into why VQRound counts so heavily on rounding initialization?

4) What is the performance ceiling of VQRound?  That is, do not use vectorization to compress the auxiliary matrices.

[1] Hu, Xing, et al. "Ostquant: Refining large language model quantization with orthogonal and scaling transformations for better distribution fitting." ICLR 2025.

[2]  Sun, Yuxuan, et al. "Flatquant: Flatness matters for llm quantization." ICML2025.

---

### Official Review · Reviewer_V8hy · 2025-10-28

**Soundness:** 2
**Presentation:** 2
**Contribution:** 2
**Rating:** 4
**Confidence:** 4

**Summary:**

This article divides the rounding matrix of quantization errors into residual subvectors and proposes a VQ reparameterization-based rounding method called VQRound. By using codebooks, it solves the problem of the huge number of parameters in the traditional Adaround method and achieves better results than GPTQ.

**Strengths:**

1. This method innovatively expresses the coefficients of round through the reparameterization of the codebook, effectively reducing the overhead of round.
2. Using Hessian to perceive residual initialization is beneficial for accelerating training convergence.

**Weaknesses:**

1. This method is an improvement of adaround; however, no comparison has been made in terms of accuracy, which limits its...

2. Both VQ-round and GPTQ can be regarded as quantization methods, which is similar to the relationship between Adaround and OBQ. However, in Table 5, VQRound+RNT is weaker than GPTQ, and furthermore, the effect when combined with GPTQ becomes even worse. These phenomena have heightened doubts about the effectiveness of VQRound.

3. Global fine-tuning increases memory overhead, which may be the reason why the performance on larger models such as 70B has not been reported.

4. There is also a risk of overfitting. For example, in Table 4, the ppl of global fine-tuning is even lower than that of fp16, but the acc in Table 5 fails to exceed that of floating-point.

5. It fails to make comparisons with cutting-edge algorithms such as QuaRot, SpinQuant, OSTQuant, and FlatQuant.

**Questions:**

1. Can you provide ablation studies on codebook size, annealing strategy, etc.?
2.See WeakNess

---

### Official Review · Reviewer_E73v · 2025-10-29

**Soundness:** 2
**Presentation:** 2
**Contribution:** 2
**Rating:** 2
**Confidence:** 4

**Summary:**

The paper presents a PTQ method called VQRound, which re-parameterizes the rounding matrix via VQ into a compact codebook, to significantly reduce the number of learnable parameters. Beyond conventional layer- or block-wise reconstruction, the authors introduce a lightweight end-to-end fine-tuning pipeline that requires a small number of samples and enables global optimization of codebooks across all layers.

**Strengths:**

1) The authors validate the effective of VQRound for various quantization configurations from 4-bit to 2-bit.
2) They theoretically demonstrate why the vectorized re-parameterization can outperform other re-parametrization techniques.
3) In contrast to layer-wise or block-wise optimizations, they implement VQRound’s end-to-end fine-tuning process.

**Weaknesses:**

1. As mentioned in Abstract, the initialization of the rounding matrix is critical for VQRound. In Table 6, however, the authors look into the initialization of the rounding matrix only for OPT models, which are undertrained and thus easy to quantize [1]. To strengthen that VQ is superior to other methods such as SVD and Kronecker product decomposition, it would be necessary to investigate the initialization of the rounding matrix for recent LLMs such as Llama 3.2 or Qwen3.
2. The baselines (RTN and GPTQ) are too limited to verify the efficacy of VQRound. To show that VQRound’s end-to-end fine-tuning process can be better than conventional layer-wise or block-wise optimization, it would be required to compare VQRound with block-wise quantization methods like OmniQuant [1].
3. Either PPL or CSR accuracy is reported. So, it is hard to determine whether VQRound performs well or not. It would be more beneficial if the authors also explore more challenging tasks (e.g., MMLU) and generation tasks (e.g., IFEval, GSM8K) with recent LLMs such as Llama 3.2 or Qwen3.
4. There are many distillation losses, but there is no ablation study about why the authors choose KL, which makes the novelty of VQRound seem marginal. It would be more beneficial if the authors try other distillation losses such as Jensen-Shannon Divergence (JSD).

[1] OmniQuant: Omnidirectionally Calibrated Quantization for Large Language Models, ICLR 2024.

**Questions:**

In Table 4, VQRound+RTN sometimes works better than VQRound+GPTQ, but there is no explanation why this happens.

---

### Official Review · Reviewer_74XB · 2025-10-31

**Soundness:** 2
**Presentation:** 2
**Contribution:** 2
**Rating:** 4
**Confidence:** 4

**Summary:**

The paper uses codebooks to guide the rounding phenomenon in quantization algorithms. This is in contrast to alternate method of adaptive rounding which employ a residual matrix that is added to the weights and is often in low-rank format.

**Strengths:**

- The empirical results look good.

**Weaknesses:**

- I strongly suggest the authors to change the name of the paper. "Atomic" in computer science is a reserved term for operation in multi-threaded execution that employ locks.
- I understand the discussion below equation (3). However, I do not agree that controlling E_max gives a bound on E_F, it's the other way around as clearly shown by the inequalities in eq (3).
- Note that the authors have been using E_max as notation to denote an L_infinity error, but on and after line 254, they use E_infinity. Please be consistent in the notation.
- Multiple new notations are parachuted in equation (5) such as a weird inequality symbol with an tilde underneath, sigmas (which I assume denote singular values) and sigma followed by arrow followed by sigma_2 (is this a limit)?

**Questions:**

- What does the size of the balls indicate in Figure 1 left?
- Since VQ is used to guide the rounding of the weights, why not just round the weights using VQ?

---

> ### Comment · Reviewer_74XB · 2025-11-25
> **No change to the review due to the lack of rebuttal**
>
> Since there is no rebuttal, I maintain my recommendation to reject this paper.

---

### Note · Authors · 2025-12-01

I have read and agree with the venue's withdrawal policy on behalf of myself and my co-authors.